# Real-Life Impact of Glucocorticoid Treatment in COVID-19 Mortality: A Multicenter Retrospective Study

**DOI:** 10.3390/jcm10204678

**Published:** 2021-10-13

**Authors:** Ana Muñoz-Gómez, Ana Fernández-Cruz, Cristina Lavilla-Olleros, Vicente Giner-Galvañ, Cristina Ausín-García, Philip Wikman, Alejandro D. Bendala-Estrada, Juan A. Vargas, Manuel Rubio-Rivas, Jaime Laureiro, Daniel Fernández-Bermúdez, Verónica A. Buonaiuto, Antonio P. Arenas de Larriva, María de los Reyes Pascual-Pérez, José N. Alcalá-Pedrajas, Ane Labirua-Iturburu Ruiz, Almudena Hernández-Milián, Marta Gómez del Mazo, Beatriz Antequera, Carmen Mella-Pérez, María de la Sierra Navas-Alcántara, Juan F. Soto-Delgado, Rosa M. Gámez-Mancera, Cristina Sardiña-González, Héctor Meijide-Míguez, José M. Ramos-Rincón, Ricardo Gómez-Huelgas

**Affiliations:** 1Internal Medicine Department, Hospital de Parla, 28981 Madrid, Spain; amg.sevilla@gmail.com; 2Infectious Diseases Unit, Internal Medicine Department, Hospital Universitario Puerta de Hierro-Majadahonda, Instituto de Investigación Sanitaria Puerta de Hierro—Segovia de Arana, 28222 Madrid, Spain; 3Internal Medicine Department, Hospital General Universitario Gregorio Marañón, 28007 Madrid, Spain; cristina.lavilla@salud.madrid.org (C.L.-O.); crisau3@hotmail.com (C.A.-G.); alejandro.bendala@gmail.com (A.D.B.-E.); 4Internal Medicine Department, Hospital Clínico Universitario de Sant Joan d’Alacant, Fundación para el Fomento de la Investigación Sanitaria y Biomédica en la Comunidad Valenciana (FISABIO), 03550 Alicante, Spain; ginervicgal@gmail.com (V.G.-G.); wikman.philip@gmail.com (P.W.); 5Internal Medicine Department, Hospital Universitario Puerta de Hierro-Majadahonda, Instituto de Investigación Sanitaria Puerta de Hierro—Segovia de Arana, 28222 Madrid, Spain; juanantonio.vargas@salud.madrid.org; 6Department of Internal Medicine, Bellvitge University Hospital, L’Hospitalet de Llobregat, 08907 Barcelona, Spain; mrubio@bellvitgehospital.cat; 7Doce de Octubre University Hospital, 28041 Madrid, Spain; jaimelaureiro@gmail.com; 8Internal Medicine Department, Costa del Sol Hospital, 29603 Marbella, Spain; danifb1911@gmail.com; 9Internal Medicine Department, Regional University Hospital of Málaga, Biomedical Research Institute of Málaga (IBIMA), University of Málaga (UMA), 29010 Málaga, Spain; drveanbu@gmail.com (V.A.B.); ricardogomezhuelgas@hotmail.com (R.G.-H.); 10Lipids and Atherosclerosis Unit, Department of Internal Medicine, Maimonides Biomedical Research Institute of Córdoba (IMIBIC), Reina Sofia University Hospital, University of Córdoba, Av. Menendez Pidal s/n, 14004 Córdoba, Spain; antoniop.arenas.sspa@juntadeandalucia.es; 11CIBER Fisiopatología de la Obesidad y Nutrición (CIBEROBN), Instituto de Salud Carlos III (ISCIII), 28222 Madrid, Spain; 12Internal Medicine Department, University General Hospital of Elda, 03600 Alicante, Spain; cperezb@coma.es; 13Internal Medicine Department, Pozoblanco Regional Hospital, 14400 Córdoba, Spain; jnalcala58@hotmail.com; 14Santa Marina Hospital, 48004 Bilbao, Spain; aneelbire.labirua-iturbururuiz@osakidetza.eus; 15Internal Medicine Department, Son Llàtzer University Hospital, 07198 Palma de Mallorca, Spain; ahernandez4@hsll.es; 16Internal Medicine Department, San Pedro Hospital, 26006 Logroño, Spain; mgmazo@riojasalud.es; 17Internal Medicine Department, Sagunto Hospital, 46520 Valencia, Spain; beantlo@gmail.com; 18Universitary Hospital Complex of Ferrol, 15405 A Coruña, Spain; mellacarmen@gmail.com; 19Internal Medicine Department, Infanta Margarita Hospital, Cabra, 14940 Córdoba, Spain; maria.sierra.navas@gmail.com; 20Internal Medicine Department, University Hospital of Salamanca, 37007 Salamanca, Spain; jfsoto@saludcastillayleon.es; 21Internal Medicine Department, University Hospital Virgen del Rocío, 41013 Sevilla, Spain; rossa_gm17@hotmail.com; 22Internal Medicine Department, Monforte de Lemos Public Hospital, 27400 Lugo, Spain; cristina.sardina.gonzalez@sergas.es; 23Internal Medicine Department, Quironsalud Hospital, 15009 A Coruña, Spain; hector.meijide@quironsalud.es; 24Department of Clinical Medicine, Miguel Hernandez University of Elche, 03203 Alicante, Spain; jramosrincon@yahoo.es

**Keywords:** corticosteroids, SARS-CoV-2, COVID-19, mortality

## Abstract

We aimed to determine the impact of steroid use in COVID-19 in-hospital mortality, in a retrospective cohort study of the SEMICOVID19 database of admitted patients with SARS-CoV-2 laboratory-confirmed pneumonia from 131 Spanish hospitals. Patients treated with corticosteroids were compared to patients not treated with corticosteroids; and adjusted using a propensity-score for steroid treatment. From March–July 2020, 5.262 (35.26%) were treated with corticosteroids and 9.659 (64.73%) were not. In-hospital mortality overall was 20.50%; it was higher in patients treated with corticosteroids than in controls (28.5% versus 16.2%, OR 2.068 [95% confidence interval; 1.908 to 2.242]; *p* = 0.0001); however, when adjusting by occurrence of ARDS, mortality was significantly lower in the steroid group (43.4% versus 57.6%; OR 0.564 [95% confidence interval; 0.503 to 0.633]; *p* = 0.0001). Moreover, the greater the respiratory failure, the greater the impact on mortality of the steroid treatment. When adjusting these results including the propensity score as a covariate, in-hospital mortality remained significantly lower in the steroid group (OR 0.774 [0.660 to 0.907], *p* = 0.002). Steroid treatment reduced mortality by 24% relative to no steroid treatment (RRR 0.24). These results support the use of glucocorticoids in COVID-19 in this subgroup of patients.

## 1. Introduction

As of July 1st, there had been 249,659 SARS-CoV-2 infections diagnosed in Spain, causing at least 28,363 deaths during the same period [1], and no effective therapy available. During the first wave of the SARS-CoV-2 pandemic, the use of corticosteroids for the treatment of COVID-19 was controversial, based on data from influenza pneumonia and SARS-CoV-1 infection [2]. However, they were used in many cases, due to the lack of effective therapy for this entity. There was not enough knowledge about the different phases of the illness nor had there been time to generate scientific evidence on the safety and effectiveness of different potential treatments [3].

Initially, drugs with presumed antiviral properties, such as hydroxychloroquine or lopinavir/ritonavir, were widely administered, but failed to demonstrate benefits [4]. Progressively, it became clear that elevation of inflammatory biomarkers and certain clinical features such as fever, pointed towards a hyper-inflammatory state. Based on this rationale, despite controversies, and in the absence of a better option, steroid treatment was considered to treat patients with COVID19 in many cases.

The first studies that analyzed the role of steroid treatment in COVID-19 did not support their use, in some cases for fear of delayed viral clearance based on the previous scarce evidence with SARS and MERS [5,6]. However, when viral clearance of SARS-CoV-2 was analyzed in patients who received corticosteroid treatment, there was no evidence of delayed viral clearance [7,8]. Furthermore, other single center studies suggested a positive impact of corticosteroids in mortality of patients with COVID-19 who needed supplementary oxygen [9,10].

The results of the RECOVERY trial [11] suggest that treatment with dexamethasone is beneficial for patients with COVID-19 that present with hypoxemia, which motivated the change in the World Health Organization recommendations [12]. In the present study, we analyze real-world data with the aim of confirming the consistency of these findings in a large Spanish multicenter cohort.

## 2. Material/Patients and Methods

### 2.1. Setting and Study Design

The present observational study is based on the SEMI (Spanish Society of Internal Medicine)-COVID-19 Registry. This is an ongoing, nationwide multicenter anonymized online database of consecutive adult patients admitted with SARS-CoV-2 laboratory-confirmed pneumonia from 131 different Spanish hospitals.

As described elsewhere [13], inclusion criteria for the registry were age ≥ 18 years and first hospital discharge with a confirmed diagnosis of COVID-19. Exclusion criteria were subsequent admissions of the same patient and denial or withdrawal of informed consent. Patients were cared for at their attending physician’s discretion, according to local protocols and clinical judgment.

A retrospective cohort study was designed to compare the differences in all-cause mortality between patients who received corticosteroids as part of the treatment for COVID-19 (independently of the dose, duration of the treatment and moment of initiation) and those who did not.

### 2.2. Data Collection

The registry includes epidemiological, clinical, laboratory and radiologic data extracted from electronic medical records. More in-depth information on the registry is available in previously published works [13].

### 2.3. Definitions

We considered SARS-CoV-2 infected patients those with a microbiological confirmation by reverse transcription polymerase chain reaction (RT-PCR) testing of a respiratory sample. Exposure to corticosteroids was defined as the use of systemic corticosteroids at any time during the hospital admission. Different types of corticosteroids were considered (dexamethasone, methylprednisolone, prednisone). The effect of different doses or different corticosteroids is analyzed elsewhere (Lavilla-Olleros C. et al. in press).

The main outcome variable was in-hospital mortality.

The diagnosis and grading of ARDS was determined according to modified Berlin criteria [14] (in non-ventilated patients, the PEEP value in the modified criteria was not taken into consideration): mild: PaO_2_/FiO_2_ 200–300 mmHg (PEEP or CPAP ≥ 5 cmH_2_O, or non-ventilated); moderate: PaO_2_/FiO_2_ 100–200 (PEEP ≥ 5 cmH_2_O, or non-ventilated); severe: PaO_2_/FiO_2_ ≤ 100 mmHg (PEEP ≥ 5 cmH_2_O, or non-ventilated).

### 2.4. Statistical Analysis

Quantitative variables were expressed as means and standard deviations (SD) and/or medians and interquartile ranges, and qualitative variables as frequencies and percentages. The association of baseline characteristics among the steroid and control cohorts with mortality was assessed through univariable conditional logistic regression to compute crude odds ratios (ORs) and their 95% confidence intervals (CIs).

To compare differences between survivors and non-survivors, the Mann–Whitney U test, X^2^ test, Fisher’s exact test or Student *t* test were used where appropriate. To explore risk factors associated with in-hospital death, univariable analysis and multivariable logistic regression models were used. Variables with a *p* < 0.05 in univariable analysis were selected into the multivariable.

To reduce the effect of corticosteroid treatment selection bias and potential confounding, we adjusted for differences in baseline characteristics by a propensity score, which predicts the patient’s probability of being treated with corticosteroids regardless of confounding factors, using multivariable logistic regression. Potential confounders considered in propensity score matching analysis were those variables included in the final model by means of stepwise backward elimination procedures. The effect of steroid treatment on clinical outcome was analyzed by a multivariable logistic regression, adjusted for major variables associated with mortality; the individual propensity score was incorporated into the model as a covariate, to calculate the propensity adjusted odds ratio (OR), as described elsewhere [10].

All statistical analyses were performed using SPSS system (version 26.0 for Windows, SPSS Inc., Chicago, IL, USA). The statistical significance level was set at a two-sided *p* value of <0.05. An odds ratio (OR) was reported along with 95% confidence interval (CI).

## 3. Results

As of July 2020, 14,921 patients were included in the SEMI-COVID-19 Registry. Among them, 5262 (35.26%) patients were treated with corticosteroids and 9659 (64.73%) patients were not (from now on, control cohort). Median time to corticosteroid treatment from symptoms onset was 10 days (IQR 7 to 14 days).

### 3.1. Clinical Characteristics

The clinical characteristics of the steroid and control cohorts are shown in Table 1. There were significant differences between the group of patients who received steroid treatment and those who did not in all the baseline characteristics and laboratory findings considered relevant (including known poor prognosis risk factors such as hypertension, obesity, diabetes, D-dimer or C-RP which were more prevalent or presented higher levels among those who received steroids), except for the total lymphocyte count, which showed no significant differences.

### 3.2. In-Hospital Mortality of Patients Treated with Corticosteroids Compared to Patients Not Treated with Corticosteroids

Overall, in-hospital mortality was 20.5%. Characteristics of survivors and non-survivors are shown in Table 2.

Notably, whereas in the univariable analysis, in-hospital mortality was higher in patients treated with corticosteroids than in controls (16.2% versus 28.5%; OR 2.068 (95% confidence interval, 1.908 to 2.242); *p* = 0.0001), when adjusting by occurrence of established ARDS (all severities included), mortality was significantly lower in the steroid treatment group (43.4% versus 57.6%; OR 0.564 (95% confidence interval, 0.503 to 0.633); *p* = 0.0001). Furthermore, we could see a greater benefit of steroid treatment in mortality as the severity of the lung damage increased (6% vs. 10.1% in mild ARDS, 26% vs. 51.2% in moderate ARDS, and 66.6% vs. 84.3% in severe ARDS). Mortality was higher in the subgroup of patients without ARDS who received steroid treatment (5% vs. 9.9%).

Age-adjusted Charlson comorbidity index (OR 1.481 (1.440–1.524), *p* = 0.000), low SpO_2_/FiO_2_ ratio at admission (OR 0.997 (0.996–0.998), *p* = 0.000), established ARDS at any point during the course of the disease 4.233 (3.958 to 4.527), *p* = 0.000), high values of LDH (OR 1.001 (1.000–1.001), *p* = 0.000) and high values of C-reactive protein (OR 1.002 (1.001 to 1.003), *p* = 0.000) were independent risk factors for mortality, whereas steroid treatment (OR 0.793 (0.7682–0.923), *p* = 0.003), Tocilizumab treatment (OR 0.458 (0.371 to 0.564), *p* = 0.001), Hydroxychloroquine treatment (OR 0.342( 0.281 to 0.416), *p* = 0.000) and Lopinavir/Ritonavir treatment (OR 0.620 (0.533 to 0.723), *p* = 0.000) were independent protective factors.

A propensity score for steroid treatment was developed to adjust for differences in baseline characteristics. The propensity score for steroid treatment included the following variables: hypertension, age-adjusted Charlson comorbidity index, diabetes, onco-hematologic underlying disease, acute coronary syndrome, COPD, obesity, moderate-severe chronic kidney disease, SpO_2_/FiO_2_, lymphocyte, LDH and C-RP at admission. The propensity-score-matched cohort is displayed in Table 1.

Table 3 shows the risk factors for mortality in both univariable and multivariable analyses, including those adjusted by the propensity score for steroid treatment. When adjusting these results including the propensity score as a covariate, the overall in-hospital mortality remained significantly lower in the steroid treatment group (OR 0.774 (0.660 to 0.907), *p* = 0.002). Steroid treatment reduced mortality by 24% relative to no steroid treatment (RRR 0.24).

## 4. Discussion

In the present study, we observed that patients with SARS-COV-2 pneumonia with established acute respiratory distress syndrome who received corticosteroids as part of their treatment had 24% inferior in-hospital mortality than those who did not.

Although retrospective studies on steroid effect on COVID-19 mortality are discordant, our results are consistent with those reported in the RECOVERY trial. Other retrospective studies support these results [9,10].

Interestingly, as reported in the RECOVERY trial, steroid treatment did not show a beneficial effect on patients who did not require oxygen, on the contrary, mortality in this subset was higher than in the non-steroid treatment group [11]. This can be at least partly explained by the characteristics of the patients who received steroids, who presented a higher prevalence of dismal prognostic factors such as hypertension, obesity or diabetes among others and higher levels of biomarkers associated with poor outcome in COVID-19 (D-dimer, C-RP). Nevertheless, we cannot exclude that in some cases steroid could have been administered too early, during the viral phase of the disease, which has been associated with worse outcomes [15,16]. In addition, corticosteroids are not exempt of adverse effects, in particular when given at high-dose, and in patients without severe involvement the risk-benefit ratio may not be favorable.

Furthermore, our results suggest that the greater the respiratory failure, the larger the impact on mortality of the steroid treatment. These results are in accordance with the physiopathology of ARDS in COVID-19 which is based in the exuberant inflammatory response in the lungs [17].

The only corticosteroid tested in the RECOVERY trial was dexamethasone, at 6 mg per day [11]. In the present study, different corticosteroids at different dosage regimens were used. This real-life use of diverse corticosteroids in COVID-19 was associated with a decrease in mortality in patients with COVID-19 that fulfilled ARDS criteria. Dexamethasone doses in the RECOVERY trial are equivalent to medium prednisone doses. It is a matter of controversy whether higher doses or even pulses would present better results in COVID-19 (Lavilla-Olleros C et al. in press) [18]. However, some authors believe that viral clearance could be affected by high dose corticosteroids, but not by low-dose [19]; consequently, this must be taken into consideration when analyzing different doses of steroid treatment for COVID-19.

The major strength of the present study was the great number of patients included, and being a multicenter registry with a highly heterogeneous population (as there were no exclusion criteria), which we believe provides real-life results, generalizable to other populations.

Limitations of the present study are those typical of retrospective studies, including the occasional missing data. Unfortunately, the precise moment of developing ARDS criteria relative to symptoms onset or corticosteroid administration was not appropriately registered, as was neither the exact corticosteroid used in each case, limiting the subgroup analysis. To overcome selection bias, we developed a propensity score for steroid treatment from the baseline characteristics of patients, and performed an adjusted analysis.

Our real-life results support evidence from clinical trials in favor of the use of steroid treatment in patients with SARS-COV-2 pneumonia and established ARDS. The ideal dosage and the optimal moment in the course of the disease require further investigation. Future studies should address as well whether steroid treatment has a class effect in COVID-19 or is specific for some drugs, and determine if synergies with other anti-inflammatory or antiviral drugs exist.

## Figures and Tables

**Table 1 jcm-10-04678-t001:** Baseline characteristics of patients in the steroid and non-steroid complete cohort and in the propensity-matched cohort.

	Complete Cohort			Propensity Score Matched Cohort		
	Non-Steroid Treatment	Steroid Treatment	*p* Value	Non-Steroid Treatment	Steroid Treatment	*p* Value
Baseline Characteristics and Underlying Medical Conditions						
Sex (Men, %)	5208 (54.0%)	3323 (63.2%)	0.000	1269 (55.1%)	1011 (65.4%)	0.000
Age (mean, SD)	65	70	0.000	67.8	70.5	0.000
Age-adjusted Charlson comorbidity index (mean)	3.4	4.0	0.000	3.7	4.2	0.000
High blood pressure	4590 (47.6%)	2983 (56.8%)	0.000	1242 (53.9%)	953 (61.5%)	0.000
Diabetes	1724 (17.9%)	1140 (21.7%)	0.000	471 (20.4%)	370 (23.9%)	0.011
Acute coronary syndrome	703 (7.3%)	485 (9.2%)	0.000	193 (8.4%)	162 (10.5%)	0.028
Moderate to severe chronic kidney disease	516 (5.4%)	388 (7.4%)	0.000	154 (6.7%)	123 (7.9%)	0.137
Dyslipidemia	3630 (37.7%)	2272 (43.3%)	0.000	964 (41.8%)	711 (45.9%)	0.012
COPD	505 (5.2%)	516 (9.8%)	0.000	170 (7.4%)	199 (12.8%)	0.000
Stroke	664 (6.9%)	417 (7.9%)	0.019	68 (3.0%)	43 (2.8%)	0.750
Obesity	1657 (18.8%)	1209 (25.3%)	0.000	543 (23.5%)	464 (30.0%)	0.000
Leukemia	93 (1.0%)	86 (1.6%)	0.000	24 (1.0%)	24 (1.5%)	0.163
Lymphoma	120 (1.2%)	92 (1.7)	0.013	32 (1.4%)	27 (1.7%)	0.378
Hematologic disease	211 (2.2%)	175 (3.3%)	0.000	55 (2.4%)	50 (3.2%)	0.116
Solid tumor with metastases	218 (2.3%)	105 (2.0%)	0.291	60 (2.6%)	34 (2.2%)	0.421
Solid tumor without metastases	581 (6.0%)	351 (6.6%)	0.119	150 (6.5%)	104 (6.7%)	0.797
Malignancy	787 (8.2%)	454 (8.6%)	0.323	208 (9%)	137 (8.8%)	0.848
Moderate to severe liver disease	90 (0.9%)	58 (1.1%)	0.315	24 (1%)	21 (1.4%)	0.372
Laboratory findings at admission						
Urea (mg/dL)	45.5	53.4	0.000	49.1	55.7	0.000
LDH (U/L)	354	403	0.000	385.58	437.4	0.000
C-reactive protein (mg/L)	52.1	70.5	0.000	94.1	130.9	0.000
Lymphocytes (per 10^6^/L)	1437.3	1169.4	0.000	1150.6	1047.4	0.090
D-Dimer (ng/mL)	1607.3	2430.9	0.000	1881.8	2563.4	0.076
Interleukin-6 (pg/mL)	47.06	95.7	0.000	53.8	103.5	0.003
Ferritin (mcg/L)	750.5	1209.2	0.000	801.4	1281.4	0.000
Respiratory parameters						
SpO_2_/FiO_2_ at admission	273.0	203.8	0.000	405.4	375.6	0.000
ARDS	1082 (11.2%)	1549 (29.6%)	0.000	597 (26.0%)	928 (60.1%)	0.000

**Table 2 jcm-10-04678-t002:** Baseline characteristics of patients in the survivor and non-survivor cohort.

	Survivors (*n* = 11,862)	Non—Survivors (*n* = 3059)	*p* Value
Baseline Characteristics and Underlying Medical Conditions			
Sex (Men, %)	6619 (55.9%)	1912 (62.6%)	0.000
Age (mean, SD)	64.18 (15.9%)	79.67 (10.6%)	0.000
Age-adjusted Charlson comorbidity index (mean)	3.08 (2.5%)	5.70 (2.41%)	0.000
High blood pressure	5423 (45.8%)	2150 (70.4%)	0.000
Diabetes	1993 (16.9%)	871 (28.6%)	0.000
Ischemic heart disease	751 (6.3%)	437 (14.3%)	0.000
Moderate to severe chronic kidney disease	517 (4.4%)	387 (12.7%)	0.000
Dyslipidemia	4360 (36.8%)	1542 (50.6%)	0.000
COPD	645 (5.4%)	376 (12.3%)	0.000
CVA	657 (5.6%)	424 (13.9%)	0.000
Obesity	2253 (20.7%)	613 (21.4%)	0.045
Onco-hematologic	1050 (8.9%)	524 (17.2%)	0.000
Hematologic disease	231 (2%)	155 (5.1%)	0.000
Leukemia	106 (0.9%)	73 (2.4%)	0.000
Lymphoma	126 (1.1%)	86 (2.8%)	0.000
Solid tumor with metastases	226 (1.9%)	97 (3.2%)	0.000
Solid tumor without metastases	633 (5.3%)	299 (9.8%)	0.000
Moderate to severe liver disease	105 (0.9%)	43 (1.4%)	0.009
Laboratory findings at admission			
Urea (mg/dL)	41.65 (29.7)	74.46 (51.35)	0.000
Lactate dehydrogenase (U/L)	347.06 (193.9)	476.80 (299.77)	0.000
C-reactive protein (mg/L)	77.04 (81.5)	127.24 (104.14)	0.000
Lymphocyte count (per 10^6^/L)	1174.68 (1773.5)	1139.90 (3317.10)	0.578
D-Dimer (ng/mL)	1623.3	6074.50	0.000
Interleukin-6 (pg/mL)	60.50 (173.8)	126.45 (183.48)	0.000
Ferritin (mcg/L)	892.35 (1039.4)	1235.03 (1375.82)	0.000
Respiratory parameters			
SatO_2_/FiO_2_ (SD)	423.73 (63.6)	356.01 (106.1)	0.000
ARDS	2483 (21.0%)	2413 (79.8%)	0.00
Treatments received			
Steroid treatment	3763 (31.7%)	1499 (49.0%)	0.000
Hydroxychloroquine	10,487 (88.5%)	2285 (74.7%)	0.000
Lopinavir/Ritonavir	7549 (63.7%)	1599 (52.4%)	0.000
Interferon	1125 (9.5%)	536 (17.6%)	0.000
Tocilizumab	948 (8.0%)	309 (10.1%)	0.000
Anakinra	64 (0.5%)	26 (0.9%)	0.046

**Table 3 jcm-10-04678-t003:** Risk factors for mortality in both univariable and multivariable analyses (including a multivariable analysis including the propensity score for steroid treatment as a covariate).

	Univariable Analysis		Multivariable Analysis		Multivariable Adjusted with Propensity Score	
N = 14.921	OR (95% CI)	*p* Value	OR (95% CI)	*p* Value	OR (95% CI)	*p* Value
Basal characteristics and underlying medical conditions						
Sex (Women)	0.756 (0.697–0.821)	0.000				
Age	1.088 (1.084–1.092)	0.000				
Age-adjusted Charlson comorbidity index	1.445 (1.420–1.470)	0.000	1.481 (1.440–1.524)	0.000	1.440 (1.397–1.485)	0.000
High blood pressure	2.824 (2.592–3.076)	0.000				
Diabetes	1.972 (1.799–2.163)	0.000				
Ischemic heart disease	2.470 (2.179–2.800)	0.000				
Moderate to severe chronic kidney disease	3.184 (2.772–3.656)	0.000				
Dyslipidemia	1758 (1.623–1.905)	0.000				
COPD	2.441 (2.135–2.791)	0.000				
Ictus	2.754 (2.420–3.315)	0.000				
Obesity	1.133 (1.024–1.254)	0.016				
Onco-hematologic	2.128 (1.900–2.384)	0.000				
Hematologic disease	2.687 (2.183–3.306)	0.000				
Leukemia	2.713 (2.008–3.665)	0.000				
Lymphoma	2.693 (2.042–3.553)	0.000				
Solid tumor with metastases	1.688 (1.326–2.148)	0.000				
Solid tumor without metastases	1.922 (1.664–2.219)	0.000				
Moderate to severe liver disease	1.596 (1.117–2.282)	0.010				
Laboratory findings at admission						
Urea (mg/dL)	1.022 (1.021–1.023)	0.000				
LDH (U/L)	1.003 (1.0020–1.003)	0.000	1.001 (1.000–1.001)	0.000	1.000 (1.000–1.001)	0.005
C-reactive protein (mg/L)	1.006 (1.005–1.006)	0.000	1.002 (1.001–1.003)	0.000	1.000 (0.999–1.002)	0.526
Lymphocytes (per 10^6^/L)	1.00 (1.00–1.00)	0.436				
D-Dimer (ng/mL)	1.00 (1.00–1.00)	0.000				
Interleukin-6 (pg/mL)	1.001 (1.001–1.002)	0.000				
Ferritin (mcg/L)	1.000 (1.000–1.000)	0.000				
Respiratory parameters						
SatO_2_/FiO_2_ (SD)	0.991 (0.991–0.992)	0.000	0.997 (0.996–0.998)	0.000	0.998 (0.997–0.999)	0.000
ARDS	14.845 (13.444–16.393)	0.000	4.233 (3.958–4.527)	0.000	4.299 (3.393–4.540)	0.000
Treatments administered						
Steroid treatment	2.068 (1.908–2.242)	0.000	0.793 (0.682–0.923)	0.003	0.774 (0.660–0.907)	0.002
Hydroxychloroquine	0.386 (0.349–0.426)	0.000	0.342 (0.281–0.416)	0.000		
Lopinavir/Ritonavir	0.626 (0.578–0.679)	0.000	0.620 (0.533–0.723)	0.000	0.636 (0.542–0.747)	0.000
Interferon	2.031 (1.816–2.271)	0.000				
Tocilizumab	1.294 (1.130–1.481)	0.000	0.458 (0.371–0.564)	0.001	0.476 (0.383–0.592)	0.000
Anakinra	1.587 (1.004–2.507)	0.048				

## Data Availability

The datasets generated during and/or analyzed during the current study are available from the corresponding author on reasonable request.

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
