# Peer review of "Real-Life Impact of Glucocorticoid Treatment in COVID-19 Mortality: A Multicenter Retrospective Study"

_jcm, 2021, doi:10.3390/jcm10204678_

Round 1

Reviewer 1 Report

This manuscript by Munoz et al., represents important insights in to the treatment of COVID-19 disease with corticosteroids. These findings have important implications of the treatment with SARS-CoV-2 infections, and have the capacity to benefit many patients with the ongoing pandemic. The authors do however, need to address minor issues in the manuscript.

General

  1. Check spelling and grammar.
  2. Ensure that SARS-CoV-2 (e.g., line 60) and COVID-19 (e.g., line 76) nomenclature is consistent.
  3. Change “2020th” to “2020”.
  4. Include units for all statistics including IQR e.g., line 163 change to (IQR 7 to 14 years).
  5. Change commas to full stops in the p values.

Introduction

  1. The authors should describe the number of cases, deaths etc as a consequence of SARS-CoV-2 to put this study in to context.
  2. The authors should include a statement of what the study will address so that the scene is set for the rest of the manuscript.

Results

  1. Explain how the propensity score was developed in this section as this should be mentioned up front, as it is utilised throughout the study.
  2. Table 1: You show lymphocyte numbers for both groups. Is their any difference between B. T or NK cells as this is important as these populations produce IL-6.
  3. Lines 164-169: Authors mention that there were significant differences between the group of patients who received steroid treatment. This should be briefly described here.

Discussion:

  1. Authors should give some insight in to the future studies that should be undertaken, and how the findings of this study can inform treatment of patients with COVID-19 disease.

Author Response

Thank you for your comments that we believe will improve our article.

Reviewer 1

Comments and Suggestions for Authors

This manuscript by Munoz et al., represents important insights in to the treatment of COVID-19 disease with corticosteroids. These findings have important implications of the treatment with SARS-CoV-2 infections, and have the capacity to benefit many patients with the ongoing pandemic. The authors do however, need to address minor issues in the manuscript.

General

Check spelling and grammar: this has been done.

Ensure that SARS-CoV-2 (e.g., line 60) and COVID-19 (e.g., line 76) nomenclature is consistent.

We have double checked that “SARS-CoV-2” is used as causative agent, and “COVID-19” is used as disease. For the sake of clarity, we have eliminated “COVID-19 pneumonia” as it could be considered redundant.

Change “2020th” to “2020”.

This has been corrected

Include units for all statistics including IQR e.g., line 163 change to (IQR 7 to 14 years).

This has been amended.

Change commas to full stops in the p values.

This has been corrected.

Introduction

The authors should describe the number of cases, deaths etc as a consequence of SARS-CoV-2 to put this study in to context.

As of July 1st, there had been 249,659 SARS-CoV-2 infection diagnosed cases in Spain, causing at least 28,363 deaths during the same period.(https://es.statista.com/estadisticas/1107506/covid-19-casos-confirmados-muertes-y-recuperados-por-dia-espana/).This information has been added to the manuscript.

The authors should include a statement of what the study will address so that the scene is set for the rest of the manuscript.

The study intends to confirm in a real-life setting the results of the clinical trials that suggest a beneficial effect in COVID-19. This is stated in the last paragraph of the introduction: “In the present study, we analyze real-world data with the aim of confirming the consistency of these findings in a large Spanish multicenter cohort”.

Results

Explain how the propensity score was developed in this section as this should be mentioned up front, as it is utilised throughout the study.

The propensity score was developed based on significant differences in baseline characteristics of patients according to exposure to steroids, analyzed by means of multivariable logistic regression analysis. This is explained in the methods section. Nevertheless, an additional sentence to introduce the propensity score in the results section has been added for clarification.

Table 1: You show lymphocyte numbers for both groups. Is their any difference between B. Tor NK cells as this is important as these populations produce IL-6.

Unfortunately, data about lymphocyte subtypes or NK cells were not registered in this database. As a large multicenter, country-wide study, we believe it is not possible to retrieve those data retrospectively.

Lines 164-169: Authors mention that there were significant differences between the group of patients who received steroid treatment. This should be briefly described here.

A sentence specifying that known poor prognostic factors suchashypertension, obesity, diabetes, D-dimer or C-reactive protein which were more prevalent or presented higher levels among patients who received steroidshas been added.

Discussion:

Authors should give some insight in to the future studies that should be undertaken, and how the findings of this study can inform treatment of patients with COVID-19 disease.

The last paragraph of the discussion has been completed with some more suggestions for future studies.

Reviewer 2 Report

  1. line 119-121, numbers in formula should be subscript;
  2. line 160, 2020th needs to be checked.
  3. Table 2, survivors (n=11.862), suppose to be n=11,862.

Author Response

Comments and Suggestions for Authors

line 119-121, numbers in formula should be subscript;

This has been corrected.

line 160, 2020th needs to be checked.

This has been amended.

Table 2, survivors (n=11.862), suppose to be n=11,862.

This has been corrected.

Reviewer 3 Report

They reported on the association between steroid use and mortality in COVID-19.

This is a very large cohort, and their report is worthwhile. However, for the size of the cohort, there is little discussion and little novelty. There should be additional discussion on why the prognosis is poor when steroids are used in patients without respiratory failure. For example, it has been reported that the prognosis is worse when steroids are administered before antiviral drugs (Shionoya et al. PLoS One. 2021 Sep 2;16(9):e0256977.)
This study was conducted as a retrospective study, so it was not possible to standardize the steroid dosage. However, I think that the details of steroid administration (type of steroid: dexamethasone, prednisolone, or methylprednisolone, and the amount of steroids converted to dexamethasone) may be described.

Author Response

Comments and Suggestions for Authors

They reported on the association between steroid use and mortality in COVID-19.

This is a very large cohort, and their report is worthwhile. However, for the size of the cohort, there is little discussion and little novelty. There should be additional discussion on why the prognosis is poor when steroids are used in patients without respiratory failure. For example, it has been reported that the prognosis is worse when steroids are administered before antiviral drugs (Shionoya et al. PLoS One. 2021 Sep 2;16(9):e0256977.)

We have added a new paragraph to the discussion commenting on the issue of steroids in patients without respiratory failure. We thank the reviewer for suggesting the recent work by Shionoya et al that we have included among the references.

This study was conducted as a retrospective study, so it was not possible to standardize the steroid dosage. However, I think that the details of steroid administration (type of steroid: dexamethasone, prednisolone, or methylprednisolone, and the amount of steroids converted to dexamethasone) may be described.

Another article by the same group discussing the dosage of steroids used in this population is currently under review (Lavilla-Olleros C. et al), which is referenced in the manuscript.

Round 2

Reviewer 3 Report

They are honestly answering my question.
If the reference number is sorted in the order in which it appeared in the paper, it is considered publishable in the present form.

Author Response

Thank you for your comments.

The reference ordder has been double checked.